# Estrogen Receptor β4 Regulates Chemotherapy Resistance and Induces Cancer Stem Cells in Triple Negative Breast Cancer

**DOI:** 10.3390/ijms24065867

**Published:** 2023-03-20

**Authors:** Ayesha Bano, Jessica H. Stevens, Paulomi S. Modi, Jan-Åke Gustafsson, Anders M. Strom

**Affiliations:** 1Center for Nuclear Receptors and Cell Signaling, Department of Biology and Biochemistry, Science & Engineering Research Center, University of Houston, Houston, TX 77204, USA; 2UT Health San Antonio, San Antonio, TX 78229, USA; 3Department of BioSciences and Nutrition, Karolinska Institutet, 171 77 Huddinge, Sweden

**Keywords:** estrogen receptor beta, estrogen receptor beta isoforms, triple negative breast cancer, chemo resistance, stemness of cancer cells, drug transporters, BCSC population, Hypoxia Inducible Factors

## Abstract

Triple Negative Breast Cancer (TNBC) has the worst prognosis among all breast cancers, and survival in patients with recurrence is rarely beyond 12 months due to acquired resistance to chemotherapy, which is the standard of care for these patients. Our hypothesis is that Estrogen Receptor β1 (ERβ1) increases response to chemotherapy but is opposed by ERβ4, which it preferentially dimerizes with. The role of ERβ1 and ERβ4 in influencing chemotherapy sensitivity has never been studied before. CRISPR/CAS9 was used to truncate ERβ1 Ligand Binding Domain (LBD) and knock down the exon unique to ERβ4. We show that the truncated ERβ1 LBD in a variety of mutant p53 TNBC cell lines, where ERβ1 ligand dependent function was inactivated, had increased resistance to Paclitaxel, whereas the ERβ4 knockdown cell line was sensitized to Paclitaxel. We further show that ERβ1 LBD truncation, as well as treatment with ERβ1 antagonist 2-phenyl-3-(4-hydroxyphenyl)-5,7-bis(trifluoromethyl)-pyrazolo[1,5-a] pyrimidine (PHTPP), leads to increase in the drug efflux transporters. Hypoxia Inducible Factors (HIFs) activate factors involved in pluripotency and regulate the stem cell phenotype, both in normal and cancer cells. Here we show that the ERβ1 and ERβ4 regulate these stem cell markers like SOX2, OCT4, and Nanog in an opposing manner; and we further show that this regulation is mediated by HIFs. We show the increase of cancer cell stemness due to ERβ1 LBD truncation is attenuated when HIF1/2α is knocked down by siRNA. Finally, we show an increase in the breast cancer stem cell population due to ERβ1 antagonist using both ALDEFLUOR^TM^ and SOX2/OCT4 response element (SORE6) reporters in SUM159 and MDA-MB-231 cell lines. Since most TNBC cancers are ERβ4 positive, while only a small proportion of TNBC patients are ERβ1 positive, we believe that simultaneous activation of ERβ1 with agonists and inactivation of ERβ4, in combination with paclitaxel, can be more efficacious and yield better outcome for chemotherapy resistant TNBC patients.

## 1. Introduction

TNBC, characterized by the lack of the three druggable targets of Estrogen Receptor α (ERα), Progesterone Receptor (PR), and Human epidermal growth factor receptor-2 (HER2), encompasses a heterogeneous group of fundamentally different diseases with different histologic, genomic, and immunologic profiles. The seminal work of Perou et al. in 2000 [1] resulted in the PAM50 intrinsic classification: Luminal A, Luminal B, HER2neu, and Basal-Like. About 71% of TNBC were found to be Basal-Like, while 77% of Basal-Like cancers were triple negative in nature [2].

TNBC patients have a very poor prognosis overall, and survival rarely extends beyond 12 months from the time of recurrence. The 5-year overall survival rate in metastatic TNBC (mTNBC) patients is 4–20%, much worse than the other types of breast cancer [3,4].

Patients with TNBC who do not achieve a pCR after neoadjuvant therapy have a significantly worse prognosis, as the chemotherapy regimen itself is responsible for altering EMT-related gene expression, and enhancing the self-renewal capacity of Breast Cancer Stem Cell (BCSC) and causing clonal selection [5,6].

Chemotherapy resistance is one of the biggest problems in treating relapsed TNBC as well as mTNBC. Survival data from clinical studies indicate that intratumoral hypoxia and increased Hypoxia Inducible Factor 1 alpha (HIF-1α) expression are associated with aggressive cancers [7,8]. Stable downregulation of HIF-1α has been shown to reverse chemotherapy resistance, inhibit proliferation, migration, and invasion of cancer cells, and slow down the tumor growth in breast cancer xenograft models [9,10].

HIF-1α has also been reported as a prerequisite for chemotherapy resistance (paclitaxel and gemcitabine) of BCSCs by inducing ROS-dependent expression of HIF-1α and HIF-2α, leading to HIF-mediated expression of IL-6, IL-8, and MDR1, thereby promoting the survival of BCSCs [10,11]. Over-expression of the multidrug resistance protein 1 (MDR1) is also associated with the resistance of taxane and anthracyclines, which are principle chemotherapeutic agents for TNBC treatment [12]. Both HIF-1α and HIF-2α are known to activate factors involved in pluripotency and regulate the stem cell phenotype, both in normal and cancer cells [8,13,14,15,16,17].

ERβ is a nuclear receptor, and it is expressed in the normal mammary gland as well as benign breast disease [18]. It is expressed in both luminal and myoepithelial cells as well as in the surrounding stromal cells [19]. ERα is found in epithelial cells in the normal mammary gland, and is abundantly expressed in Luminal A and Luminal B breast cancers [20].

Of the five alternatively spliced transcript variants of the ERβ gene ERβ1-5 [21,22], only ERβ1 and splice variants, ERβ2, ERβ4, and ERβ5, commonly known as C-terminal variants, are expressed as proteins [23].

The ERs share a conserved structural and functional domains common to all Nuclear Receptors [24]. Domains A–D are identical in ERβ1, ERβ2, ERβ4, and ERβ5; however, the E/F domains, containing the LBD, Nuclear Localization Signal (NLS), and Activation Function 2 (AF2), are uniquely truncated in ERβ2-5, and each isoform has a unique AF2 domain [25]. Thus, ERβ1 is the only receptor that has a fully functional LBD, and ERβ2 has a somewhat inefficient smaller pocket to bind ligands. ERβ4 and ERβ5 lack the critical part of the LBD, disabling them from binding ligand [23]. ERβ1 can form homo and hetero dimers with ERα, and the various ERβ isoforms. ERβ1 preferentially dimerizes with ERβ4 [25]. The ERβ isoforms ERβ2, β4, and β5 have not been observed to form homodimers [25], and it is unknown if they can act independently of ligands as monomers.

It has been shown that expression of ERβ1 mRNA in breast cancer cell lines was significantly lower than in normal breast epithelial cells, together with extensive methylation of promoter 0 N, one of the two promoters predominantly transcribing ERβ1, in breast cancer cell lines [26,27,28,29]. Analysis of TCGA data confirms the loss of ERβ1 in all breast cancer types; only about 10% of all types of breast cancer patients express ERβ1 mRNA, whereas, 90% of the breast cancer tissues express one or more of the different truncated isoforms [30].

ERβ1 is a tumor suppressor in breast cancer as well as prostate cancer [31,32]. ERβ1 opposes HIF signaling [33], while the ERβ splice variant ERβ2 stabilizes HIF factors and increase the hypoxic signaling in prostate cancer [34]. When it comes to ERβ4, there is a big knowledge gap, as it has not been well studied in breast cancer or other cancers. We have previously shown that among all ERβ isoforms, ERβ4 is the only isoform that causes mammosphere formation in MCF10A cells [32], indicating the increased frequency of cancer stem cells. The MCF10A cells with ectopic ERβ4 expression also have upregulated stem cell gene transcripts (see Appendix A). The current study explores further the role of ERβ4 in TNBC.

## 2. Results

### 2.1. ERβ Isoforms Endogenously Expressed in TNBC Cell Lines Affect Response to Chemotherapy

We used several TNBC cell lines, SUM159, MDA-MB-231, BT549, and HCC1806, to study the effect of ERβ isoforms on chemo sensitivity to Paclitaxel. We selected these TNBC cell lines with mutant TP53 [35] in order to eliminate any effect due to wild type or mutant status of the TP53 gene. We used CRISPR/Cas9 technology to manipulate endogenous expression of the ERβ1 by causing a frame shift mutation in the LBD sequence unique to ERβ1 in Exon 8, leading to early termination of the LBD, thereby disabling ERβ1 to bind E2 and initiate the gene transcription cascade. The CRISPR ERβ1 LBD truncation is referred to as CRERB1. We validated the phenotype of the CRERB1 by using 2 different siRNAs for ERβ1, as well as using an antagonist to ERβ PHTPP [36]. Figure 1 shows the localization of Guide-RNA and Dicer substrate interfering RNA (DsiRNA) in the ERβ gene.

We used lentivirus transduction of CRISPR/Cas9 with guide RNA targeting the ERβ4 5′ splice site, disrupting the ERβ4 splice variant from being transcribed, which was validated by qPCR for the exon specific to ERβ4 only (see Appendix A).

Challenge of the different cell lines with paclitaxel at different doses showed that truncation of the LBD accompanied with the disruption of the AF2 domain of ERβ1 induced resistance to cell death caused by paclitaxel. In all the 4 cell lines evaluated, MDA-MB-231, BT-549, HCC1806 and SUM159, such truncation caused an increase in resistance to the paclitaxel treatment compared to control cells (Figure 2A–C,E). While knocking down the ERβ4 increased the sensitivity of the MDA-MB-231 cells to the paclitaxel treatment, see Figure 2A. To validate that it is indeed ERβ4 that is responsible for influencing the chemo sensitivity, we created an ERβ4 overexpressing SUM159 cell line. The ERβ4 knock out and ERβ4 overexpression cell lines demonstrate opposing sensitivity to Paclitaxel; overexpression of ERβ4 in SUM159 cells caused chemotherapy resistance, similar to the CRERB1, see Figure 2D.

The clonogenic assay for SUM159, shows that there is a 10-fold increase in the fraction of SUM159 ERβ4-seeded cells that retain the capacity to produce colonies after paclitaxel treatment compared to SUM159 control (see Figure 2F).

### 2.2. Specific Truncation of Endogenous ERβ1 LBD or Treatment with the ERβ1 Antagonist PHTPP Increase Expression of ABC Transporters

We analyzed expression of the drug efflux transporters MDR1 (p-GP) and ABCG2 in MDA-MB-231, HCC1806, and SUM159 cells, as this is the most common reason of paclitaxel resistance, as well as paclitaxel being a substrate of MDR1 [37]. Truncation of ERβ1 LBD in MDA-MB-231 as well as HCC1806 showed a more than 6-fold increase of MDR1 transporters (see Figure 3A,B). Treatment with the ERβ1 antagonist PHTPP of SUM159 cells also showed a similar increase (see Figure 3C). We saw no significant change on the transporter expression level, when ERβ4 was knocked down in MDA-MB-231. However, we observed an increase in ABCG2 expression in MDA-MB-231 cells over expressing ERβ4 compared to control transduced cells, see Appendix A. For levels of ERβ4 over expression in SUM159 and MDA-MB-231, see Appendix A.

### 2.3. Truncation of Endogenous ERβ1 LBD or Treatment with the ERβ1 Antagonist PHTPP Induces Expression of Pluripotency Factors in MDA-MB-231, HCC1806, BT549, and SUM159 Cells

To determine if ERβ and its variants have any effect on the stemness of the cancer cells, we analyzed expression of Yamanaka factors OCT4, SOX2, and KLF4 that induce pluripotency in normal cells [38]. We discovered that expression of the factors was increased in HCC1806, BT549, and MDA-MB-231 cells with a truncated LBD of ERβ1 indicating the regulation of stem cell markers by ERβ1 (see Figure 4A–C). Treatment of SUM159 cells with ERβ1 antagonist PHTPP also induced c-Myc, OCT4, and ALDH1A1 see Figure 4C. Comparing truncation of ERβ1 LBD with using PHTPP in BT549 cells we found a similar regulation of Nanog, OCT4, ALDH1A1, and Vimentin (see Figure 4D,F). In MDA-MB-231, we observed that the pluripotent factors in the ERβ4 knock down line were expressed at a lower level than the control cells, indicating that ERβ4 is involved in inducing the pluripotent factors; the result is shown in Figure 4A. To further validate that specifically inhibiting ERβ1 expression increased expression of the pluripotency factor Nanog, we used siRNA directed at the ERβ1 unique LBD sequence, see Appendix A.

We also observed an upregulation of different EMT markers in the MDA-MB-231 CRERB1 cell line, see Figure 4E. Since ALDH1 is deemed a marker of Epithelial BCSCs and vimentin is a marker of mesenchymal BCSCs with increased expression in CD24^low^/CD44^high^ [39], we decided to investigate the expression level of these two markers in the various cell lines, and Figure 4F–I shows a consistent result of upregulation of Vimentin and ALDH1 in all cell lines with either CRERB1 or treatment with antagonist PHTPP, pointing to the likelihood of increased stem-like cells.

### 2.4. Increased Cancer Stem Cell Population in Cell Lines Treated with ERβ1 Antagonist

We used the SORE6-GFP reporter as an indicator of the cancer stem cells in MDA-MB-231, and compared the SORE6+ population in stable SORE6-GFP reporter cells, and the same treated with PHTPP using flow cytometry. Figure 5A shows about a 5-fold increase in the SORE6+ cells in the ERβ1 Antagonist PHTPP-treated MDA-MB-231 cell population. We also treated SUM159 cells with PHTPP at 200 nM for 24 h and performed ALDEFLUOR Assay after 48 h. The result shown in Figure 5B reveals that PHTPP treatment increased the ALDH1+ population from the non-treated cells.

Figure 5C shows the ICC of SUM159 SORE6 stable cells and SUM159 SORE6-CRERB1 stable cells. There is a higher number of cells expressing SORE6-GFP and SOX2 in CRERB1 cells, suggesting an increase in the stem-cell like population in CRERB1 cells. All the SORE6+ cells also show SOX2 staining, validating the SORE6 reporter. We also observed an increase in the SORE6+ population in the SUM159 SORE6-ERβ4 cell lines, when the SORE6-GFP cells were counted and compared against the count of the DAPI stained cells, shown in Appendix A.

In addition, we could show that expression of the transporter ABCG2 was increased in ERβ4 over expressing SUM159 cells see Appendix A.

### 2.5. Hypoxia Inducible Factors HIF-1α and HIF-2α Are Involved in Mediating the Regulation of Chemo Resistance and Cancer Stem Cell Population in TNBC

When we cultured MDA-MB-231 control and MDA-MB-231 CRERB1 cells in normoxia and hypoxia and analyzed expression of Nanog, OCT4, and SOX2, we saw the potentiation of Nanog and SOX2 in the hypoxic environment see Figure 6A. Under hypoxic conditions, HIF factors are stabilized, resulting in increased transcripts of stem cell genes. And this potentiation is observed in both Control as well as CRERB1 cells.

Next we transfected a mixture of siRNA for HIF-1/2α in the MDA-MB-231 CRERB1 cell line, which had previously shown to have elevated stem cell markers, and we observed an attenuation of the transcripts of Nanog and OCT4 in the HIF-1/2α knock down see Figure 6B. This result suggests that perhaps the HIF-1/2α play a role together with the ERβ variants in mediating the regulation of stem cells.

We next studied the impact of ERβ4 expression on HIF-1α expression, using ICC, in SUM159 cells. The comparison of SUM159 cells with those overexpressing ERβ4 show a clear increase in HIF-1α expression (see Figure 6C), which could imply stabilization of HIF-1α by ERβ4.

## 3. Discussion

Since the discovery of in of ERβ in 1996 [40], it has been widely studied in various tissues. Because of its low level of expression, it has been challenging to study the role of ERβ in Breast Cancer. In the past the focus has been on studying the effect of ERβ and its variants by using overexpression constructs in cell lines. We are of the opinion that not only does ERβ protein has a low expression, but it also has a high turnover, and that is why it is not detected easily in breast cancer cell lines. Despite the low expression level, the TCGA data, among other studies, confirms the expression of ERβ variants in all breast cancer subtypes; additionally, it confirms the loss of ERβ1 expression in over 80% of breast cancers [30].

The TNBC cell lines were chosen to study the effect of ERβ and its isoforms unfettered by the cross talk between ERα and ERβ. Of the four TNBC cell lines selected, three map to the CLDN1-low, and one maps to CLDN1-high, according to the Perou Intrinsic Basal subtype [1]. All four cell lines are p53 mutant and are further characterized under the Lehmann TNBC-subtype and the associated genetic mutations in Table 1 [1,41].

### 3.1. ERβ1 and ERβ4 Modulate the Chemotherapy Sensitivity via the Drug Efflux Transporters

When we use this ERβ1 LBD truncation strategy on four different TNBC cell lines, we observe an increased resistance to paclitaxel, and induction of stem cell factors indicating that the low level of endogenously expressed ERβ1 in TNBC cell lines has some function. To validate this ligand effect, we used the ERβ antagonist PHTPP [36] on the different TNBC cell lines and we see the same effect as truncating the ERβ1 LBD.

The truncated ERβ1 would be similar to the other splice variants, which have a truncated LBD. As the isoforms do not form dimers by themselves or with other isoforms, we expect the truncated ERβ1 should not form a dimer with ERβ4, but only with full length ERβ1. It is conceivable that the ERβ4 that does not have sufficient ERβ1 to partner with is free to act as a monomer and effect the phenotype observed.

CLDN1 sensitizes TNBC cells to chemotherapy drugs including Paclitaxel, 5-fluorouracil, and doxorubicin [42]. However, we observed the same effect of ERβ1 truncation in CLDN1-high (HCC1806) and CLDN1-low (MDA-MB-231, BT549, SUM 159) cell lines, indicating that ERβ’s role is perhaps independent of CLDN1 expression.

Induction of p53-mediated apoptosis in tumor cells is considered a desirable outcome of cancer therapy, whereas induction of cell-cycle arrest may interfere with drugs that target mitosis and reduce the efficacy of drugs like Paclitaxel [43]. A p53 mutation is expected to decrease the basal level of sensitivity to Paclitaxel. However, in all these cells lines with the p53 mutation, we observed that truncation of ERβ1 LBD, as well as overexpression of ERβ4, increased resistance to Paclitaxel; in contrast, ERβ4 knock down sensitized the cells to paclitaxel. This indicates that this opposing regulation of paclitaxel sensitivity by ERβ1 and ERβ4 occurs in spite of the p53 mutation.

In a recent study, it was shown that silencing of total ERβ in MDA-MB-231 TNBC cells expressing mutant TP53 resulted in decreased expression of TP53-target genes, leading to tumor growth and reduced apoptosis [44]. It is established that mutant Tp53 is associated with chemotherapy resistance, and that ERβ1 and its variants have opposing effect in breast and prostate cancer [31,32,34,45,46]. Our study in four different mutant Tp53 TNBC cell lines show increased chemotherapy resistance and cell survival with truncation of ERβ1 LBD. However, in the same mutant Tp53 TNBC cell lines, knockdown of ERβ4 showed increased sensitivity to paclitaxel. It would be interesting to investigate further the effect on TP53 target genes and apoptosis when either ERβ1 LBD is truncated or ERβ4 is knocked down. We hypothesize that this might also show opposing results.

### 3.2. ERβ1 and ERβ4 Regulate Stemness of Cancer Cells and Promote Epithelial to Mesenchymal Transition

It is accepted that there are several cancer stem cell (CSC) pools with distinct biological properties within individual tumors, including quiescence, drug resistance, ability to undergo epithelial-to-mesenchymal transition (EMT), invasion and metastasis, reflecting the increased levels of plasticity that are characteristic of these cells [47,48,49]. This intratumoral heterogeneity arising from the genetic or epigenetic mutational profiles and the Tumor Micro Environment (TME) is believed to govern therapy resistance, tumor progression by influencing the stemness properties of cancer cells [50].

The increased expression of some or all of the induced pluripotency factors in all the TNBC the cells studied, as a result of truncation of LBD in ERB1, and the attenuation of the same with ERβ4 knock down, suggests that ERβ1 and ERβ4 influence the stemness of the TNBC cells in opposing manner.

BCSCs from an individual tumor can exist in distinct mesenchymal-like and epithelial-like states with differences in proliferation, invasiveness, and localization within tumor mass [51]. Epithelial cells primarily employ cytokeratin in their cytoskeleton, whereas mesenchymal cells use vimentin, and carcinomas hijack this EMT program, where cytokeratin-positive epithelial cancer cells begin to express vimentin to drive invasion and metastasis [52]. Vimentin has been used as a marker for pre-metastatic cells undergoing EMT and high vimentin expression is also associated with worse prognosis [53].

The increased expression of Vimentin in the epithelial breast cancer cells, due to CRERB1 and over expression of ERβ4 alludes to the possibility of cells undergoing EMT, and increased transcripts of ALDH1 indicate increase in BCSC subpopulation. BCSCs are thought to play a major role in driving disease recurrence, due to the intrinsically enhanced therapeutic resistance that results from high expression of multidrug transporters, enhanced DNA damage checkpoint activation and repair mechanisms, and altered cell-cycle kinetics [54].

We observed the enrichment of both Mesenchymal BCSC (m-BCSC) as well as epithelial BCSC (e-BCSC) subpopulations, characterized by Vimentin and ALDH1, respectively, in MDA-MB-231, HCC1806, and BT549 when the TNBC cell lines were treated with ERβ antagonist treatment, also confirming the tumor suppressor role of ERβ1 in mammary epithelial cells, reported by many researchers [55].

### 3.3. Increased Frequency of BCSC Population with ERβ1 Antagonist Treatment

BCSCs were first described in 2003, based on CD44 and CD24 expression [56]. Later, in 2007, it was discovered that a subpopulation of mammary cells with high aldehyde dehydrogenase (ALDH) activity could initiate tumors in vivo and in vitro [57]. Considering the heterogeneity of CSCs, including a quiescent mesenchymal population and a proliferative epithelial population with metastatic ability, different markers are required to enrich for and identify the different cancer stem cells. ALDH1+ and CD44^+^/CD24^low^ are considered the gold standard for enrichment of BCSCs. [58]. The two markers ALDH and Vimentin have been used to identify the two different subsets of breast BCSC [59]. The m-BCSCs (CD44^+^/CD24^−^) have a slow proliferating, quiescent phenotype and express Vimentin, whereas the e-BCSC (ALDH+) are more proliferative [39].

The stem cell phenotype in embryonic stem cells (ESCs) is maintained by a triad of master transcriptional regulators, OCT4, SOX2, and NANOG, which promote stemness by upregulating genes involved in pluripotency and self-renewal while suppressing genes involved in differentiation [38,60]. SORE6 reporter identifies the subpopulation of tumor cells that are quiescent and enriched for Stem Cell in many solid tumors [48,61,62,63].

The CSC state is dynamic, and cancer cells can transition between stem and differentiated states in response to chemotherapy, radiotherapy, or other stimuli within the TME [64,65,66]. Non-CSCs can activate their self-renewal program and convert from SORE6 negative to positive cells with the latter having a higher self-renewal capacity.

The ALDEFLUOR assay identifies the ALDH+ epithelial cancer stem cells, which have been shown to enrich for the BCSCs, with increased tumorigenicity [57]. Single-cell gene expression analysis has revealed that in the ALDH+ population, at least two CSC subpopulations exist, one proliferative and one quiescent, indicating the plasticity of the CSC [67].

Both these reporters were used to analyze the putative cancer stem cell population in SUM159 and MDA-MB-231 cell lines using flow cytometry. Treatment with ERβ1 antagonist led to a 5–9-fold increase in the cancer stem cell population identified using both markers, see Figure 5A,B.

### 3.4. HIFs Mediate the Cancer Cell Stemness

One major mechanism of stemness maintenance induced by hypoxia is the activation of hypoxia-inducible factors (HIFs) [13,14]. It is well established that hypoxia increases chemotherapy resistance, up-regulating the ABC transporter MDR-1 [68], and causes increase of cancer stem cell population by enhancing expression of pluripotency stem cell factors [69]. ERβ1 has previously been shown to inhibit HIF-1α expression in both breast cancer [70] and prostate cancer [71,72]. However, the ERβ variants ERβ2, and ERβ5 have been shown to stabilize HIF-1α counteracting the inhibitory effect of ERβ1 [32,34]. The increase of HIF-1α expression by ERβ4 is shown in Figure 6C, where ICC signals for HIF-1α is increased in SUM159 cells expressing ERβ4, indicating the opposing role of ERβ1 and ERβ4 in stabilizing HIF1α.

Hypoxia, through hypoxia-inducible factor (HIF), is known to induce a human Embryonic SC (hESC)-like transcriptional program, including the induced pluripotent stem cell (iPSC) inducers OCT4, NANOG, SOX2, KLF4, and cMYC, in various cancer cell lines, including breast cancer [8]. As shown in Figure 4A–D, the basal level of Nanog and Oct 4 is increased in three different CRERB1 TNBC cell lines; therefore, the MDA-MB-231 CRERB1 cells were used to evaluate the effect of HIF on these stem cell markers by knocking out both HIF-1α and HIF-2α. The results in Figure 6B suggest that HIF factors mediate the regulation of the Nanog and OCT4 mRNA by ERβ1. It has been shown that stem cell genes are indeed regulated by HIF factors [69].

## 4. Materials and Methods

### 4.1. Cell Lines and Materials

The MDA-MB-231, HCC1806, and BT549 cell lines were obtained from the American Type Culture Collection. The SUM159 cell line was obtained from Thomas C. MDA-MB-231, HCC1806, and BT549 cells were maintained in RPMI-1640 (Invitrogen Inc., Carlsbad, CA, USA) medium supplemented with 10% fetal bovine serum (FBS) (PEAK, Wellington, CO, USA), and antibiotic-antimycotic (Invitrogen Inc., Carlsbad, CA, USA). The SUM159 cell line was maintained in DMEM (Invitrogen Inc., Carlsbad, CA, USA) medium supplemented with 10% fetal bovine serum (FBS) (PEAK, Wellington, CO, USA), and anti-anti (Invitrogen Inc., Carlsbad, CA, USA). All experiments used cells below passage 30. ERβ antagonist PHTPP (Tocris, Minneapolis, MN, USA). Paclitaxel was purchased from Millipore (St. Louis, MO, USA). MTS reagent was from Biovision (Milpitas, CA, USA).

### 4.2. Preparation of RNA and qPCR

RNA extraction was performed with Qiagen mRNA extraction kit according to standard protocol. cDNA was synthesized from 1 μg of total RNA with First Strand System according to standard protocol (Invitrogen Inc. NY, USA). Real-time PCR was performed with SYBR Green I dye master mix (Applied Biosystems Foster City, CA, USA). Primers (Integrated DNA Technologies, Inc. Coralville, IA, USA) were:
36B4F, 5′—GTGTTCGACAATGGCAGCAT—3′
R, 5′—GACACCCTCCAGGAAGCGA—3′ (reference gene) ALDH1AF, 5′—GCTGGCGACAATGGAGTCAA—3′ 
R, 5′—ACGGCCCTGGATCTTGTCAG—3′OCT4F, 5′—GGATGCTGTGAGCCAAGG—3′
R, 5′—GAACAATGATGAGTGACAGACAG—3′ CAIXF, 5′—AGATGAGAAGGCAGCACAGAA—3′
R, 5′—GAAGTGGCATAATGAGCAGGA—3′ c-MycF, 5′—CTGGTGCTCCATGAGGAGAC—3′
R, 5′—CTTTTCCACAGAAACAACATC—3′ SOX2F, 5′—TTGCTGCCTCTTTAAGACTAGGA—3′ 
R, 5′—CTGGGGCTCAAACTTCTCTC—3′NanogF, 5′—ACCCAGCTGTGTGTACTCAA—3′ 
R, 5′—GGAAGAGTAAAGGCTGGGGT—3′ KLF4F, 5′—CGGACATCAACGACGTGAG—3′
R, 5′—GACGCCTTCAGCACGAACT—3′ VimentinF, 5′—TCGTTTCGAGGTTTTCGCGTT—3′ 
R, 5′—CGACTAAAACTCGACCGACTC—3′ MDR-1F, 5′—AGGAAGCCAATGCCTATGACTTTA—3′ 
R, 5′—CAACTGGGCCCCTCTCTCTC—3′ABCG2F, 5′—TTTCCAAGCGTTCATTCAAAAA—3′ 
R, 5′—TACGACTGTGACAATGATCTGAGC—3′ ERβ4F, 5′—ACTTGCTGAACGCCGTGACC—3′ 
R, 5′—TTTTCTCCCCATCTCGCATGC—3′ FibronectinF, 5′—AAACTTGCATCTGGAGGCAAACCC—3′
R, 5′—AGCTCTGATCAGCATGGACCACTTN-CadherinF, 5′—GGTGGAGGAGAAGAAGACCAG—3′
R,5′—GGCATCAGGCTCCACAGT—3′EpCAMF, 5′—CTGCCAAATGTTTGGTGATG—3′
R, 5′—ACGCGTTGTGATCTCCTTCT—3′FOXC2F, 5′—CCGTCTCGGAAGCAGCAT—3′
R, 5′—TGAGCGCGATGTAGCTGTAG—3′

7500 Fast Real-Time PCR System (Applied Biosystems) using optimized conditions for SYBRGreen I dye system: 50 °C for 2 min, 95 °C for 10 min, followed by 40–50 cycles at 95 °C for 15 s and 60 °C for 50 s. Optimum primer concentration was determined in preliminary experiments, and amplification specificity confirmed by dissociation curve. Each experiment was carried out in triplicates and the housekeeping control gene, 36B4, was used for normalization of target gene abundance. Data was analyzed by the comparative 2∆∆CT method [73].

### 4.3. CRISPR/Cas9 Manipulation of Cell Lines to Cause LBD Truncation of Endogenous ERβ1 and Mutation of ERβ4 5′ of ERβ4 Unique Exon Splice Site

MDA-MB-231, BT549, HCC1806, and SUM159 cells were infected with the lentivirus Lenti CRISPR-V2-Puro (Addgene plasmid #98280), non-target or containing guideRNA specific for human ERβ1 LBD and ERβ4 splice site at 2 m.o.i (multiples of infection). The control in all experiments is cells infected with non-targeting CRISPR virus vector.

Annealed oligos for guide-RNA specific for truncation of ERβ1 LBD and knock down of ERβ4 was cloned into the Esp3I site of the lentivirus vector pLentiCRISPR-V2 puro. Primers (Integrated DNA Technologies, Inc. Coralville, IA, USA) were:
ERβ LBD truncationF, 5—CACCGCAACATGAAGTGCAAAAATG—3′
R, 5—AAACCATTTTTGCACTTCATGTTGC—3′ERβ4 knock downF, 5′—CACCGCTTTTCTCCCCATCTGTAA—3′ 
R, 5′—AAACTTACAGATGGGGAGAAAAGC—3′ControlF, 5′—CACCGTATTACTGATATTGGTGGG—3′
R, 5′—AAACCCCACCAATATCAGTAATAC—3′

### 4.4. DsiRNA for ERβ1

The Double stranded interfering RNA were obtained from IDT, and cells were transfected using Lipofectamine 2000

DsiRNA ERβ1 Sequence 1

rGrGrC rArUrG rGrArA rCrArU rCrUrG rCrUrC rArArC rArUG A

DsiRNA ERβ1 Sequence 2

rUrCrA rUrGrU rUrGrA rGrCrA rGrArU rGrUrU rCrCrA rUrGrC rCrCrU

### 4.5. ERβ4 Ectopic Expression

We cloned the 3 x FLAG tagged ERβ4 variant into lenti6-V5-D-TOPO (Invitrogen Thermo Fisher) and infected the SUM159 cells with virus at 2 m.o.i (multiple of infection). The cells were selected with 5 μg/mL of blasticidine for at least one week before analysis.

### 4.6. Cytotoxicity MTS Assay

3000–5000 cells/well were seeded in a final volume of 100 µL into a 96 well plate in RPMI medium with 10% FBS and left to attach overnight. On day 2, the media was replaced in all wells and the cells were treated with Paclitaxel at a range of 1–100 nM. A positive control was left untreated to determine absorption for 100% viability, and a negative control after Triton (10%) treatment. The cells were incubated for 48 h. On day 4, 10 µL MTS reagent (BioVision) was added in each well and the cells were incubated. The absorption was measured at 490 nm after 90-120 min.

### 4.7. Clonogenic Survival Assay

500–1000 cells were seeded in 6 well plates, in triplicate, and then treated with 4 nM Paclitaxel for 48 h. Subsequently the media was changed and colonies were allowed to form over 12 days. The survival fraction was calculated after normalizing for the plating efficiency, as described earlier [74].

### 4.8. Immuno Cyto Chemistry

Cells were seeded at a density of 2.5 × 10^4^ per well in the 4-well Nunc Lab-Tek II CC Chamber Slide, and left overnight. The cells were fixed using 4% paraformaldehyde in PBS pH 7.4 for 10 min at room temperature. After washing 3 × in PBS, the samples were incubated in 1% Triton x-100. The cells were then washed 3 × in PBS, and blocked for an hour in 3% BSA in PBS. Cells were then incubated in the diluted primary antibody in 3% BSA at room temp for 1 h on a shaker. After washing the cells 3 × with PBS for 5 min, the cells were incubated in the secondary antibody diluted in 1% BSA in the dark, for 1 h. After washing the cells, the slide was mounted using mounting medium with DAPI. For multicolor staining, the primary and secondary antibodies were incubated simultaneously. The slides were imaged using Keyence BZ-X810.

Antibodies used were SOX2 (Abcam #181616) and HIF1α (Cell Signaling #14179). Secondary antibodies AlexaFluor 488, and 594 were obtained from abcam. The cells were counted using the Keyence BZ-X810 Analyzer

### 4.9. FACS Analysis

SORE6 transduced cells control and treated with PHTPP for 48 h, were collected by trypsinization, washed with PBS and resuspended in PBS with 5% FBS, at a dilution of 1 × 10^6^ cells/mL. The cells were analyzed using Fortressa X-20. Non transduced wild type cells were used as negative control, and GFP expressing cells were used as positive control for gating purposes.

### 4.10. Aldefluor Assay

The ALDEFLUOR^TM^ assay kit (StemCell Technologies, Durham, NC, USA) was used to analyze the population with a high ALDH enzymatic activity using the manufacturers protocol. Trypsinized cells, 1 × 10^6^ cells/mL, were suspended in ALDEFLUOR assay buffer containing ALDH substrate (5 μL BAAA/mL of ALDEFLUOR buffer) and incubated for 40 min at 37 °C. As negative control, for each sample of cells an aliquot was treated with diethylaminobenzaldehyde (DEAB), a specific ALDH inhibitor, (5 μL DEAB/mL of ALDEFLUOR buffer). The sorting gates were established using AU565 cells as positive control. Analysis of the samples were done on Fortressa X-20.

### 4.11. Statistics

The Prism Graph 9.1.4 was used to analyze and graph the data. The error bars are expressed as the mean ± s.e.m. An unpaired two-tailed *t*-test was used to compare the differences between two groups with welch. The significance is presented as * *p* < 0.05, ** *p* < 0.005, and *** *p* < 0.001, **** *p* < 0.0001, and non-significant differences are presented as ns.

## 5. Conclusions

There are currently no targeted therapies for TNBC. ERβ4 being a driver for chemo sensitivity, tumorigenicity and EMT, can be a potential drug target for patients expressing ERβ4. Although the targeting of ERβ as a therapy for various cancers including Breast cancer has been conceptualized over the last couple of decades, since ERβ1 has been acknowledged as a tumor suppressor, but the focus has always been on developing an agonist to activate its function [55]. A recent clinical trial (NCT02352025) showed inhibition of proliferation in patients with TNBC as measured by a decrease in Ki-67 with exposure to S-equol, a novel ERβ agonist, with potential immune activation [75].

This study identifies ERβ4 as a potential druggable target for TNBC patients, the majority of whom express ERβ4. Simultaneous activation of ERβ1 with agonists and inactivation of ERβ4, in combination with paclitaxel, can be more efficacious and yield better outcomes for chemo-resistant TNBC patients. The regulation of chemotherapy sensitivity for other chemo drugs should also be investigated, as well as the effect in WT TP53 TNBC cell lines.

## Figures and Tables

**Figure 1 ijms-24-05867-f001:**
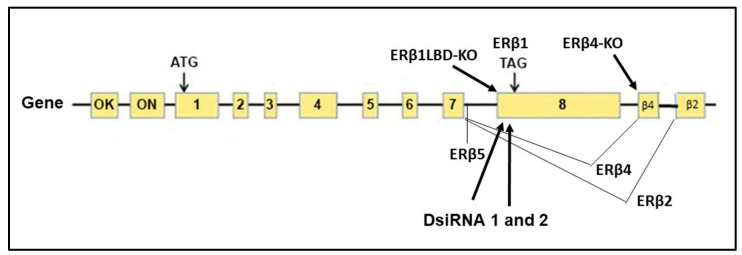
Schematic overview of the ERβ gene, two promoters 0 K and 0 N, localization of guideRNA for truncation of ERβ1 LBD, guideRNA for mutation of ERβ4 specific splice site and localization of the two DsiRNA oligos.

**Figure 2 ijms-24-05867-f002:**
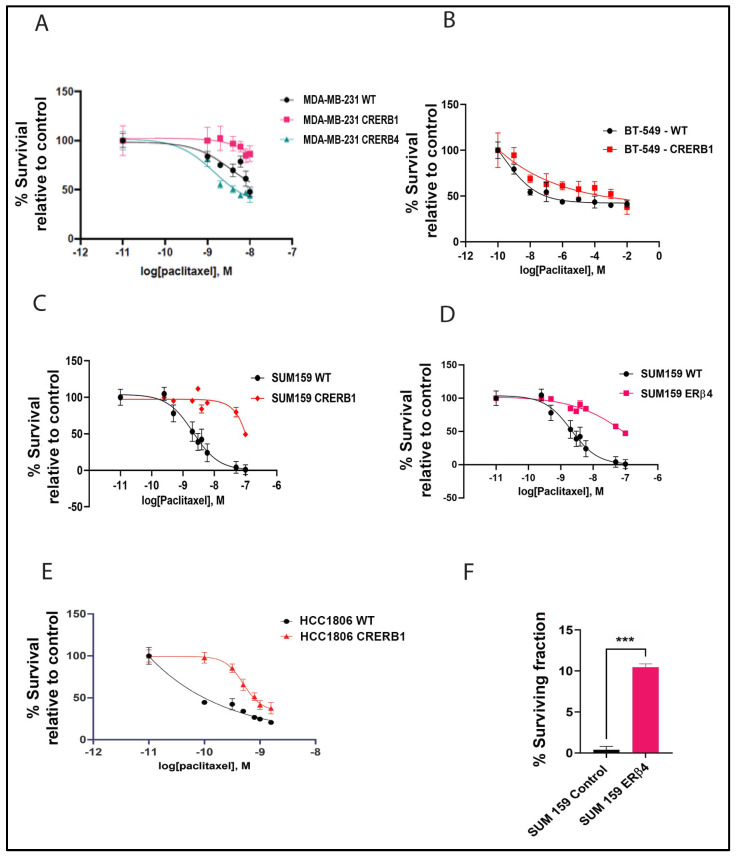
Chemotherapy treatment of TNBC cell lines shows opposing effect of ERβ1 and ERβ4. (**A**). MTS assay comparing Log dose response curves for Paclitaxel in MDA-MB-231 WT, CRERB1 and CRERB4. (**B**). Log dose response curves for paclitaxel in BT549 WT compared to CRERB1 (**C**). Log dose response curves for paclitaxel in SUM159 WT compared to CRERB1. (**D**). Log dose response curves for paclitaxel in SUM159 WT compared to ERβ4 expressing cells. (**E**). Log dose response curves for paclitaxel in HCC1806 WT compared to CRERB1. (**F**). SUM159 and SUM159 ERβ4 cells were seeded at 500 cells/well in triplicate, and treated with paclitaxel for 48 h. Colonies were then allowed to grow in regular media for 12 days. The surviving fraction was calculated after normalizing for plating efficiency. Two-tailed Student’s *t*-test was used *** *p* < 0.001.

**Figure 3 ijms-24-05867-f003:**
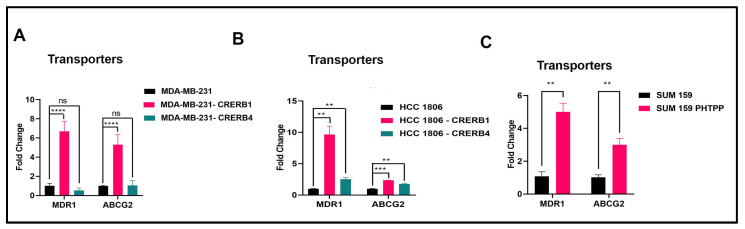
ABC transporters are regulated by ERβ1. (**A**). Quantitative real-time PCR analysis (qRT-PCR) of MDR1 and ABCG2 mRNA levels normalized to 36B4 in MDA-MB-231, MDA-MB-231 CRERB1, and MDA-MB-231 CRERB4 cell lines. Transporters are upregulated by truncating the ERβ1 LBD in MDA-MB-231 cells, while deleting the ERβ4 exon has no effect on transporter expression. (**B**). qRT-PCR analysis of transporters in HCC1806. MDR1 and to a lesser extent ABCG2 is upregulated in HCC1806 cells by ERβ1 LBD truncation while deleting the ERβ4 exon causes a slight increase in the transporter expression. (**C**). qPCR analysis of transporters in SUM159 WT and SUM159 treated with the ERβ antagonist PHTPP, show upregulation of both MDR-1 and ABCG2. All results are shown as mean ± s.e.m. from three technical replicates. Data was analyzed using unpaired Two-tailed Student’s *t*-test. *p*-values: ** *p* < 0.01, *** *p* < 0.001, **** *p* < 0.0001 and ns = non-significant.

**Figure 4 ijms-24-05867-f004:**
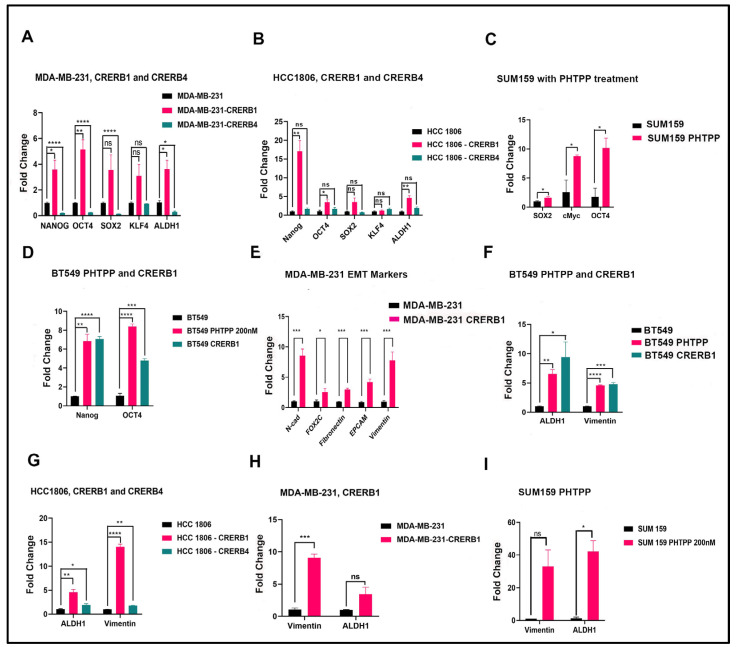
Pluripotency stem cell markers, EMT markers, Vimentin and ALDHA1 are regulated by ERβ isoforms. (**A**). qPCR analysis of stem cell markers in MDA-MB-231 cells WT, CRERB1, and CRERB4, (**B**). qPCR analysis of stem cell markers in HCC1806 cells WT, CRERB1, and CRERB4. (**C**). qPCR analysis of stem cell markers in SUM159 cells and PHTPP-treated. (**D**). qPCR analysis of stem cell markers in BT549 cells WT, PHTPP-treated, and CRERB1. (**E**). qPCR analysis of EMT markers in MDA-MB-231 WT and CRERB1. (**F**). qPCR analysis of Vimentin and ALDH1 in BT549 WT, PHTPP, and CRERB1. (**G**). qPCR analysis of Vimentin and ALDH1 in HCC1806 WT, CRERB1, and CRERB4. (**H**). qPCR analysis of Vimentin and ALDH1 in MDA-MB-231 WT and CRERB1. (**I**). qPCR analysis of Vimentin and ALDH1 in SUM159 WT and PHTPP-treated. All results are shown as mean ± s.e.m. from three technical replicates. Data was analyzed using unpaired Two-tailed Student’s *t*-test. *p*-values: * *p* < 0.05, ** *p* < 0.01, *** *p* < 0.001, **** *p* < 0.0001 and ns = non-significant.

**Figure 5 ijms-24-05867-f005:**
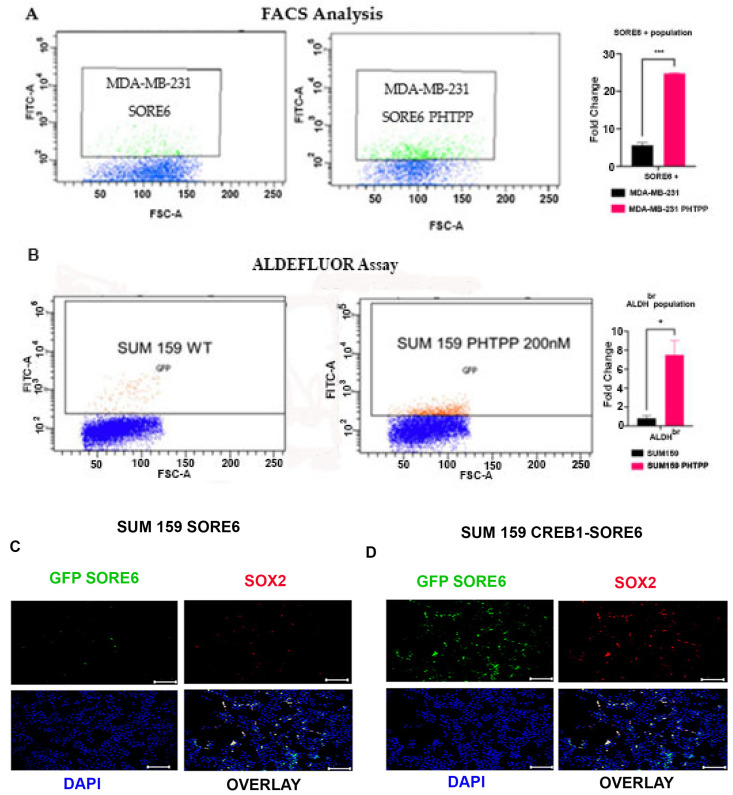
(**A**). MDA-MB-231 cells with integrated SORE6-GFP cancer stem cell reporter was treated with vehicle or 200 nM PHTPP for 48 h and then subjected to FACS analysis. The fold change is based on three technical replicates, and *** *p* < 0.001. (**B**). SUM159 cells were treated with vehicle or 200 nM PHTPP for 48 h and then subjected to ALDEFLUOR assay, followed by FACS analysis. The fold change is based on three technical replicates, and * *p* < 0.05. (**C**,**D**). WT and CRERB1 SUM159 cells with integrated SORE6-GFP reporter were analyzed using GFP for activation of SORE6, and Immunofluorescence for SOX2 scale bars are 200 μm.

**Figure 6 ijms-24-05867-f006:**
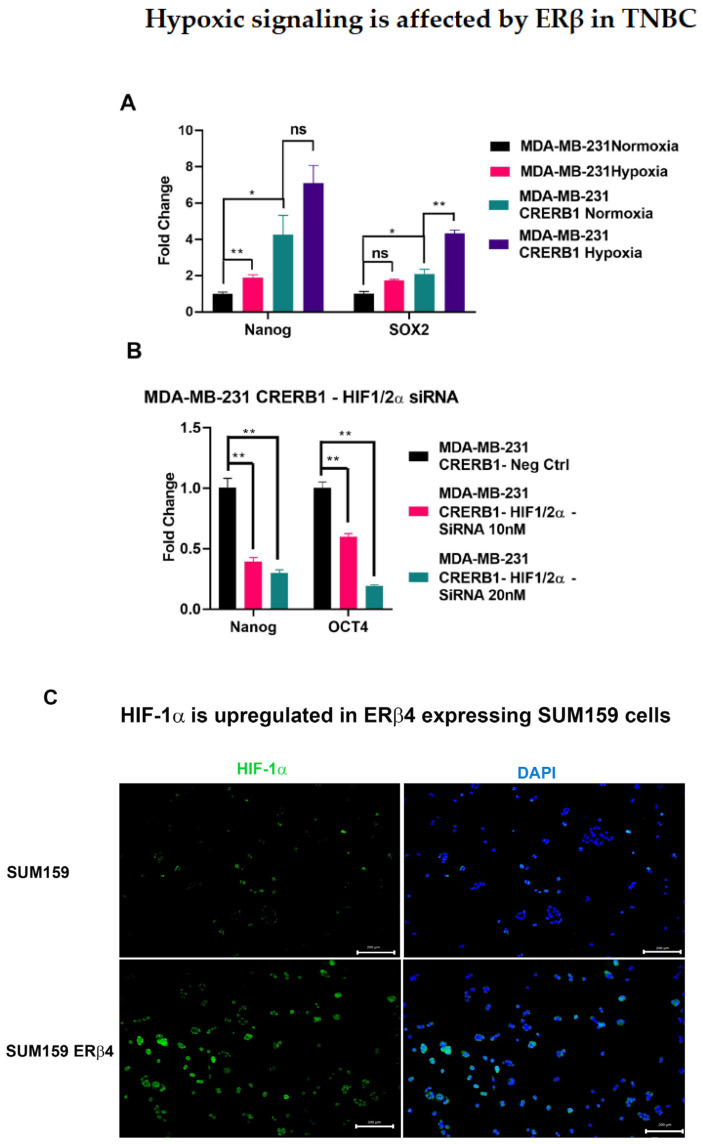
(**A**). qRT-PCR analysis of stem cell markers Nanog, Oct4 and SOX2 expression levels after subjecting MDA-MB-231 WT or CRERB1 cells to normoxia or hypoxia (1% O_2_ for 4 h). All results are shown as mean ± s.e.m. from three technical replicates. Data was analyzed using unpaired Two-tailed Student’s *t*-test. *p*-values: * *p* < 0.05, ** *p* < 0.01 and ns = non-significant. (**B**). MDA-MB-231 CRERB1 cells transfected with control siRNA and siRNA to both HIF-1α and HIF-2α and analyzed for expression of Nanog and Oct4 after 48 h). All results are shown as mean ± s.e.m. from three technical replicates. Data was analyzed using unpaired Two-tailed Student’s *t*-test. *p*-values: ** *p* < 0.01. (**C**). SUM159 cells overexpressing ERβ4 show increased HIF-1α expression as analyzed by Immunofluoroscence. Scale bars are 200 μm.

**Table 1 ijms-24-05867-t001:** Characterization of Cell Lines used in this study [1,35,41].

Lehmann TNBC Subtype [41]	Genetic Abnormalities [41]	Cell Line [35]	CLDN-1 Status	Gene Mutations [35]
Basal-like 2	Growth factor-signaling pathways (EGFR, MET, NGF, Wnt/β-catenin, IGF-1R), Glycolysis, gluconeogenesis, Expression of myoepithelial markers	HCC1806	High	TP53; CDKN2A; UTX
Mesenchymal-like	Cell motility, Cell differentiation Growth factor signaling EMT	BT-549	Low	TP53; PTEN; RB1
Mesenchymal stem-like	Similar to M above + Low proliferation, Angiogenesis genes	SUM159PT	Low	TP53; PIK3CA
		MDA-MB-231	Low	TP53; BRAF; CDKN2A; KRAS; NF2; PDGFRA

Assignment of Cell lines to the Lehmann-TNBC-subtype, CLDN-1 status, genetic abnormalities and specific genetic mutations based on gene expression data for TNBC cell lines.

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
