# Peer review of "Estrogen Receptor β4 Regulates Chemotherapy Resistance and Induces Cancer Stem Cells in Triple Negative Breast Cancer"

_ijms, 2023, doi:10.3390/ijms24065867_

Round 1

Reviewer 1 Report

In this interesting study Bano et al aimed to study the role of ERbeta1 and ERbeta4 in the chemoresistance and stem cell phenotype in TNBC. To this end they used 4 different TNBC cell lines that were subjected to CRISPR/Cas9 gene editing and subsequent analysis. The data presented in the paper are interesting and could have clinical value in the treatment of drug-resistant TNBC in the future. 

However, I have some major concerns and would request the authors to address the following comments before I can recommend this article for publications:

Introduction:

1. Referring to lines 52-58 in the introduction, the paragraph covers how different types of treatments has impacted various subtypes of TNBC. However, the references in this paragraph are not focused on the Lehman subtypes, but TNBC in general. Therefore, it is not correct to refer treatment response of different types of breast cancer therapy to specific subtypes, unless the authors can provide literature supporting that statement specifically. 

2. Also, in the introduction in line 77-78 there is a statement that ERbeta is the dominant ER in the normal mammary gland as well as in benign breast disease. Could the authors elaborate on this point since traditionally ERalfa is considered to be the primary Estrogen receptor in the breast. The citation regarding this is quite dated and has only one figure on this subject and in addition does not cover benign disease. Since this statement goes against the general dogma it would need additional (and more recent) supporting literature. On this point it seems contradictory that in the discussion the authors state that generally the expression of Erbeta is low (line 265). I would also like to note that a simple search in the protein atlas indicates that ERalfa has much higher expression (in most luminal cells) than Er beta in the normal breast gland. 

Regarding this paragraph I also cannot find supporting literature in the to the statement in line 79 that ERa is only found in 10-15% of epithelial cells in normal mammary gland.  

Results:

3. This study is based on cell line models where CRISPR editing was used to modulate the expression of different ERbeta isoforms in four cell lines: SUM159, MDA-MB-231, BT549 and HCC1806. However, the effect of the CRISPR editing on ERbeta expression was not directly measured with RNA or protein expression. This is a crucial experiment for this paper. The authors absolutely need to be confirm the effect of the gene editing in each cell line. Since the cells were not single cell cloned (and subsequently sequenced) the level of ERbeta knock down in the cells will rely entirely on the efficient of the CRISPR/Cas system. This is likely to vary between cell lines and gRNAs, and could change over time since this CRISPR method will essentially create a pool of WT+edited cells that could have different growth rate for example. As the study focuses on the differences between the expression of ERbeta1 and ERbeta4 the experiments that are needed to characterize the CRISPR cell lines are: 1) to measure and compare the expression of ERbeta1 and Erbeta4 in all WT cell lines (to show the level of expression in WT cells) 2) show and compare the expression of ERbeta1 and Erbeta4 in all CRISPR edited cell lines (since either gRNA could be affecting the expression of more than one variants of ERbeta even though the binding site of the gRNA is located at a splicing site) 3) use western blot to show that the editing of ERbeta is also translated to protein level. To be able to draw all subsequent conclusions in the paper the efficacy of the CRISPR/Cas9 editing will need to be determined using the experiments described above. 

(Importantly figure S2 shows deduction of expression of the Erbeta4 splice variant in one cell line. Here, the expression of ERbeta 1 should also be included, and for all cell lines.)

4. The sequences of the gRNA for ERbeta LBD truncation to not align with the human genome in the genome browser (when using the blat function). Could the authors clarify where it should bind exactly (the genomic coordinates). Also the sequence of the non-target control should be provided.  

5. Also, for the terminology throughout the paper, since these are not single cell clones, it is not correct to say that ERbeta1 and ERbeta4 were knocked out. That is also evident from figure S2.

6. The level of overexpression of ERbeta4 in SUM159 (figure 2F) should be evaluated. And since overexpression of ERbeta4 has the same effect on drug response as knock down of ERbeta1 it would be very interesting to see if ERbeta1 expression is altered in the overexpressing cell lines. In addition, did the authors consider checking the expression of ABC transporters in ERbeta4 over-expressing cell line?

7. The authors state that PHTPP is a ERbeta1 antagonist (line 160). Has its selectivity of ERbeta1 over ERbeta4 been established, and could the authors provide a literature or experiments supporting this? Could it be that in all PHTPP experiments there is inhibition of both isoforms?

8. Since the authors clearly have HCC 41806 – ERbeta4 cell line (figure 3) adding that line to figure 2 and assess the chemo sensitivity would really improve that figure. 

9. In line 184-185 the authors state that the upregulation of NAOG, OCT4, SOX2, KLF4 and ALDH1 in cells expressing truncated LBD of ERbeta1 confirms the tumor suppressor role of ERbeta1. Could the authors clarify why they think that increased expression of stem cell markers after ERbeta1 downregulation confirms that ERbeta1 is a tumor suppressor? These genes are traditionally considered to be markers of stemness rather then oncogenic potential of cells.

10. In figure 5C the authors claim that all the SORE6+ cells also show SOX2 staining. This is not clear from the figures (but all cells that express SOX2 appear to be also GFP positive). Also, the scale bar is missing in figure 5C and D. From the DAPI pictures the images appear not to be in the same magnification. This needs to be corrected. For figure S4 the IF images should be provided in the manuscript (as was done with figure 5C and 5D).

11. In figure 6B the level of HIF1/2alfa knock down following siRNA treatment needs to be established, for example with qPCR. Also, assessing the expression of ERbeta 1 and ERbeta4 after HIF1/2alfa knock down would be entirely appropriate to support the connection between HIF-1/2alfa and ERbeta variants (as stated in line 246). 

Also, are the authors suggesting there is a synergistic effect here? Can they please elaborate on that point. I would rather think that the results show that both ERbeta and HIF1/2 alfa affect the expression of Nanog and SOX2, but how or if they work together is unclear.

12. The experiments done to produce supplemental figure 3A and B are not described in the main text, nor is it clear what they are confirming (as stated in line 120).

13. In table one HCC1806 is attributed to the Lehmnan TNBC subtype “basal like 2” but in line 165 it is a stated that HCC1806 is a basal like 1 subtype. This need to be clarified. 

Discussion:

14. In the discussion: in line 392 the authors claim that “As shown in Figure 4 A-D, the basal level of Nanog and Oct 4 is increased in 4 different CRERB1 TNBC cell lines” But There are only three CRERB1 cell lines there – SUM159 CRERB1 was not used in this figure (but it would be much better to include it).

15. In the last lines of the discussion the authors write that the results from figure 6B suggests that HIF factors mediate the regulation of Nanog and OCT4 by ERbeta1. Could the authors clarify this statement? I would think that figure 6B shows that HIF1alfa regulates these factors regardless of ERbeta1, since their expression goes down after HIF1alfa knock down (regardless of the Erbeta1 downregulation).

16. Regarding title of the paper, I think that the phrase that a factor “induces Cancer Stem Cells” is not correct, rather one should say that a factor “induces the formation of Cancer Stem Cells” or something to that effect. Also, why did the authors put the focus on ERbeta4 in the title when the focus of the paper is primarily on ERbeta1 (most of the CRISPR cells are ERbeta1 knock out and PHTPP is a ERbeta1 antagonist according to the authors).

Materials and methods:

17. Please provide the manufacturer of the lentiviruses used in the study.

18. Scale bars for fig 5 and 6 are missing.

19. The name “Anti-anti” (line 404 and 406) is probably  not correct (is it antibiotic-antimicotic?).

20. I can find any description on the concentration of the PHTPP that was used or how long the cells were treated in figure 3.

21. The sequences for the siRNA HIF1/2alfa are missing (used in figure 6B). Did the authors use siRNAs for both HIF1 and HIF2 simultaneously? Also a more detailed description of the siRNAs used for ERbeta1 should be provided in M&Ms.

Other comments:

22. The title of figure 6 is not included. Unless it is meant to be the sentence that is above fig6c which would mean that it is not in a correct place (and not very descriptive of the overall figure).

23. As a pure suggestion,  I do think that it is very beneficial for the reader to have table 1 displayed much earlier in the manuscript to understand the difference between the different cell lines used in the study. 

Author Response

Thank you for good suggestions our response is below in italic

Comments and Suggestions for Authors

In this interesting study Bano et al aimed to study the role of ERbeta1 and ERbeta4 in the chemo resistance and stem cell phenotype in TNBC. To this end, they used four different TNBC cell lines that were subjected to CRISPR/Cas9 gene editing and subsequent analysis. The data presented in the paper are interesting and could have clinical value in the treatment of drug-resistant TNBC in the future. 

However, I have some major concerns and would request the authors to address the following comments before I can recommend this article for publications:

Introduction:

  1. Referring to lines 52-58 in the introduction, the paragraph covers how different types of treatments has impacted various subtypes of TNBC. However, the references in this paragraph are not focused on the Lehman subtypes, but TNBC in general. Therefore, it is not correct to refer treatment response of different types of breast cancer therapy to specific subtypes, unless the authors can provide literature supporting that statement specifically. This statement is removed.
  2. Also, in the introduction in line 77-78 there is a statement that ERbeta is the dominant ER in the normal mammary gland as well as in benign breast disease. Could the authors elaborate on this point since traditionally ERalfa is considered the primary Estrogen receptor in the breast. The citation regarding this is quite dated and has only one figure on this subject and in addition does not cover benign disease. Since this statement goes against the general dogma, it would need additional (and more recent) supporting literature. On this point it seems contradictory that in the discussion the authors state that generally the expression of ERbeta is low (line 265). I would also like to note that a simple search in the protein atlas indicates that ERalfa has much higher expression (in most luminal cells) than ER beta in the normal breast gland. We have added a new reference and removed the old one. The statement in line 265 (268 revised version) is referring to expression of ERbeta in breast cancer not normal breast. The expression of ERbeta1 is reduced in breast cancer compared to normal breast. However, our findings is that even this remaining low expression is still functional when we analyze its effect on the cancer stem cell population.

Regarding this paragraph I also cannot find supporting literature in the to the statement in line 79 that ERa is only found in 10-15% of epithelial cells in normal mammary gland.  We have removed this statement.

Results:

  1. This study is based on cell line models where CRISPR editing was used to modulate the expression of different ERbeta isoforms in four cell lines: SUM159, MDA-MB-231, BT549 and HCC1806. However, the effect of the CRISPR editing on ERbeta expression was not directly measured with RNA or protein expression. This is a crucial experiment for this paper. The authors absolutely need to be confirm the effect of the gene editing in each cell line. Since the cells were not single cell cloned (and subsequently sequenced) the level of ERbeta knock down in the cells will rely entirely on the efficient of the CRISPR/Cas system. This is likely to vary between cell lines and gRNAs, and could change over time since this CRISPR method will essentially create a pool of WT+edited cells that could have different growth rate for example. As the study focuses on the differences between the expression of ERbeta1 and ERbeta4 the experiments that are needed to characterize the CRISPR cell lines are: 1) to measure and compare the expression of ERbeta1 and Erbeta4 in all WT cell lines (to show the level of expression in WT cells) 2) show and compare the expression of ERbeta1 and Erbeta4 in all CRISPR edited cell lines (since either gRNA could be affecting the expression of more than one variants of ERbeta even though the binding site of the gRNA is located at a splicing site) 3) use western blot to show that the editing of ERbeta is also translated to protein level. To be able to draw all subsequent conclusions in the paper the efficacy of the CRISPR/Cas9 editing will need to be determined using the experiments described above. We are aware of the problems with this method to use the CRISPR-Cas9 technology. To address this problem we are using strong selection (the Lenti CRISPR/Cas9 constructs are selectable by puromycin) in addition, we use cells with low passage after transduction.

(Importantly figure S2 shows deduction of expression of the Erbeta4 splice variant in one cell line. Here, the expression of ERbeta 1 should also be included, and for all cell lines.) We could easy show the results for ERbeta4 since the qPCR primer was made to the unique exon of ERbeta4 that was lost when the splicing site was mutated. The situation is quite different for ERbeta1 since here we only rely on a mutation causing a frame shift thus the mRNA will be very similar to wt mRNA and is difficult to detect by qPCR.

  1. The sequences of the gRNA for ERbeta LBD truncation to not align with the human genome in the genome browser (when using the blat function). Could the authors clarify where it should bind exactly (the genomic coordinates). Also the sequence of the non-target control should be provided.  Wrong sequence given should be Fwd: CACCGCAACATGAAGTGCAAAAATG and Rev: AAACCATTTTTGCACTTCATGTTGC, (human hg19, Chr 14:64,699,984 – 64,700,035) Control Fwd: CACCGTATTACTGATATTGGTGGG and Rev: AAACCCCACCAATATCAGTAATAC
  2. Also, for the terminology throughout the paper, since these are not single cell clones, it is not correct to say that ERbeta1 and ERbeta4 were knocked out. That is also evident from figure S2. This have been changed to knockdown
  3. The level of overexpression of ERbeta4 in SUM159 (figure 2F) should be evaluated. And since overexpression of ERbeta4 has the same effect on drug response as knock down of ERbeta1 it would be very interesting to see if ERbeta1 expression is altered in the overexpressing cell lines. In addition, did the authors consider checking the expression of ABC transporters in ERbeta4 over-expressing cell line? ABCG2 transporter have been analyzed in ERbeta4 over-expressing cell line see figure S5.
  4. The authors state that PHTPP is an ERbeta1 antagonist (line 160). Has its selectivity of ERbeta1 over ERbeta4 been established, and could the authors provide a literature or experiments supporting this? Could it be that in all PHTPP experiments there is inhibition of both isoforms? ERβ4 have lost a part of the ligand-binding domain important for ligand binding so it is unlikely that PHTPP have the same effect on ERβ4 as on ERβ1.
  5. Since the authors clearly have HCC 1806 – ERbeta4 cell line (figure 3) adding that line to figure 2 and assess the chemo sensitivity would really improve that figureThis is ERβ4 knock down and not over expression.
  6. In line 184-185 the authors state that the upregulation of NANOG, OCT4, SOX2, KLF4 and ALDH1 in cells expressing truncated LBD of ERbeta1 confirms the tumor suppressor role of ERbeta1. Could the authors clarify why they think that increased expression of stem cell markers after ERbeta1 downregulation confirms that ERbeta1 is a tumor suppressor? These genes are traditionally considered markers of stemness rather than oncogenic potential of cells. We are thankful for a good point. Tumor suppressor is the wrong word. However, stemness have been shown to be important for tumor initiation and metastasis. Maybe the correct word is metastasis inhibitor.
  7. In figure 5C the authors claim that all the SORE6+ cells also show SOX2 staining. This is not clear from the figures (but all cells that express SOX2 appear to be also GFP positive). Sorry for the confusion, however, all SORE6+ cells express GFP since SORE6 is a promoter construct driving expression of GFP. Also, the scale bar is missing in figure 5C and D. From the DAPI pictures the images appear not to be in the same magnification. This needs to be corrected. For figure S4 the IF images should be provided in the manuscript (as was done with figure 5C and 5D). The images have been matched for same magnification and scale bars have been added.
  8. In figure 6B the level of HIF1/2alfa knock down following siRNA treatment needs to be established, for example with qPCR. In addition, assessing the expression of ERbeta 1 and ERbeta4 after HIF1/2alfa knock down would be entirely appropriate to support the connection between HIF-1/2alfa and ERbeta variants (as stated in line 246). Our previous findings indicate that the ERbeta variants directly interact with HIF1/2 to cause stabilization in absence of hypoxia. Thus, we do not expect any changes in mRNA or protein levels of the variants.

Also, are the authors suggesting there is a synergistic effect here? Can they please elaborate on that point? I would rather think that the results show that both ERbeta and HIF1/2 alfa affect the expression of Nanog and SOX2, but how or if they work together is unclear. Our conclusion is that HIF1/2 is necessary for ERbeta1/variants effect on Nanog and SOX2. Under normoxic conditions specifically inhibiting ERbeta1 causes Nanog and SOX2 to increase but if siRNA to HIF1/2 alpha is present under this normoxic condition the effect of inhibiting ERbeta1 is less pronounced on Nanog and SOX2. This is indicating that HIF-1/2alpha is induced by inhibiting ERbeta1 and is required for ERbeta1 inhibition to have effect on Nanog and SOX2.

  1. The experiments done to produce supplemental figure 3A and B are not described in the main text, nor is it clear what they are confirming (as stated in line 120). This experiment is just confirming the findings with ERbeta antagonist PHTPP and truncation of the ERbeta1 LBD. In S3 A and B we use siRNA specifically targeting only ERbeta1 and not ERbeta2, ERbeta4 or ERbeta5. In conclusion we have three independent methods to show that specifically targeting ERbeta1 increase Nanog expression.
  2. In table one HCC1806 is attributed to the Lehmnan TNBC subtype “basal like 2” but in line 165 it is a stated that HCC1806 is a basal like 1 subtype. This need to be clarified. This statement have been removed.

Discussion:

  1. In the discussion: in line 392 the authors claim that “As shown in Figure 4 A-D, the basal level of Nanog and Oct 4 is increased in 4 different CRERB1 TNBC cell lines” But There are only three CRERB1 cell lines there – SUM159 CRERB1 was not used in this figure (but it would be much better to include it). This is fixed.
  2. In the last lines of the discussion the authors write that the results from figure 6B suggests that HIF factors mediate the regulation of Nanog and OCT4 by ERbeta1. Could the authors clarify this statement? I would think that figure 6B shows that HIF1alfa regulates these factors regardless of ERbeta1, since their expression goes down after HIF1alfa knock down (regardless of the Erbeta1 downregulation). In figure 6B we use MDA-MB-231 CRERB1 which high a higher level of Nanog and SOX2 because of ERbeta1 inhibition. Adding siRNA to HIF-1/2 to this cell line is reducing Nanog and SOX2 expression. Our conclusion is that HIF-1/2 is needed for CREB1 to increase Nanog and SOX2.
  3. Regarding title of the paper, I think that the phrase that a factor “induces Cancer Stem Cells” is not correct, rather one should say that a factor “induces the formation of Cancer Stem Cells” or something to that effect. Also, why did the authors put the focus on ERbeta4 in the title when the focus of the paper is primarily on ERbeta1 (most of the CRISPR cells are ERbeta1 knock out and PHTPP is a ERbeta1 antagonist according to the authors). We have previously reported that ERbeta4 is frequently expressed in TNBC in agreement with other labs finding. In addition, we found that only ERbeta4 of all isoforms of ERbeta could induce mammosphere formation in MCF-10A cells. Our conclusion is that ERbeta4 is the major driver of cancer stem cell formation.

Materials and methods:

  1. Please provide the manufacturer of the lentiviruses used in the study. Invitrogen (Thermo Fisher) for over expression, and Lenti CRISPR-V2-Puro Addgene plasmid #98280, for CRISPR
  2. Scale bars for fig 5 and 6 are missing. Scale bars have been added
  3. The name “Anti-anti” (line 404 and 406) is probably not correct (is it antibiotic-antimicotic?). Yes antibiotic-antimycotic
  4. I can find any description on the concentration of the PHTPP that was used or how long the cells were treated in figure 3. Treatment with PHTPP was at 200 nM for 24 hours that we found to be optimal dose
  5. The sequences for the siRNA HIF1/2alfa are missing (used in figure 6B). Did the authors use siRNAs for both HIF1 and HIF2 simultaneously? Yes. Also a more detailed description of the siRNAs used for ERbeta1 should be provided in M&Ms. The origin of HIF siRNA was from Thermo Fisher Silence Select HIF-1alpha ID s6541, HIF-2alpha ID s4698, (Neg control #4390843). The sequence of the ERbeta1 siRNA is given in the M&M

Other comments:

  1. The title of figure 6 is not included. Unless it is meant to be the sentence that is above fig6c which would mean that it is not in a correct place (and not very descriptive of the overall figure). This is changed.
  2. As a pure suggestion, I do think that it is very beneficial for the reader to have table 1 displayed much earlier in the manuscript to understand the difference between the different cell lines used in the study. We appreciate if table 1 can stay in the discussion.

Reviewer 2 Report

Bano et al. ijms-2101694 " Estrogen receptor beta4 regulates chemotherapy resistance and induces cancer stem cells in triple negative breast cancer " is a valuable paper showing that ERβ4 has a potential druggable target for TNBC patients. Furthermore, the authors showed that activation of ERβ1 with agonists and inactivation of ERβ4 can be a better outcome for chemo-resistant TNBC patients. However, some points were difficult for the reviewer to understand. The reviewer hopes that providing more information (described below) will help improve this study's quality. 

1.     Page 3, line 105, introduction - The authors described the alteration of stem cell gene transcripts using figure S1. The reviewer believes that figure S1 is an important result in Results section 2.3. Therefore, the reviewer believes that the results in figure S1 should be moved to the Results section 2.3 to discuss the results in detail.

2.     As shown in Figure 2A, the authors showed that The ERβ4 knock out (MDA-MB-231 CRERB4) demonstrated high sensitivity to paclitaxel. If so, the reviewer believes the authors must investigate whether sensitivity to paclitaxel returns to WT in MDA-MB-231 CRERB4 with a restored expression of ERβ4

3.     Page 6, Figure 4 – In Figure 4, the genes examined in A to D are different. The reviewer thinks that the genes examined in Fig. 4A should also be applied to B to D, and future issues based on the results should be discussed in the Discussion section.

4.     Page 5, line 164- the authors described, “This could be explained by the fact that HCC1806 cell line belongs to a different Lehmann TNBC-subtype - Basal-Like 1, with different molecular signatures”. In the Lehmann TNBC subtype, HCC1806 belongs to a Basal-like 2. The reviewer believes that authors need to elaborate on why HCC1806 could belong to Basal-Like 1in the discussion section. 

Minor comment

1.     Page 1, line 15, abstract –the authors used CAS/CRSPR in this section, but the reviewer thinks the correct word is CRISPR/Cas9.

2.     Page 1, line 16, abstract –the authors used Do-main in this section, but the reviewer thinks the correct word is Domain.

Author Response

Thank you for good comments our response is below in italic

Comments and Suggestions for Authors

Bano et al. ijms-2101694 " Estrogen receptor beta4 regulates chemotherapy resistance and induces cancer stem cells in triple negative breast cancer " is a valuable paper showing that ERβ4 has a potential druggable target for TNBC patients. Furthermore, the authors showed that activation of ERβ1 with agonists and inactivation of ERβ4 can be a better outcome for chemo-resistant TNBC patients. However, some points were difficult for the reviewer to understand. The reviewer hopes that providing more information (described below) will help improve this study's quality. 

  1. Page 3, line 105, introduction - The authors described the alteration of stem cell gene transcripts using figure S1. The reviewer believes that figure S1 is an important result in Results section 2.3. Therefore, the reviewer believes that the results in figure S1 should be moved to the Results section 2.3 to discuss the results in detail. The reason for not having this figure in the main text is that the cells used is normal mammary epithelial cells and is not cancer. It is an independent confirmation of the potential of ERbeta4 as an inducer of stem cell factors.

  1. As shown in Figure 2A, the authors showed that The ERβ4 knock out (MDA-MB-231 CRERB4) demonstrated high sensitivity to paclitaxel. If so, the reviewer believes the authors must investigate whether sensitivity to paclitaxel returns to WT in MDA-MB-231 CRERB4 with a restored expression of ERβ4 We believe the results would be hard to interpret since it is very difficult to control the absolute level expressed from an expression construct delivered by lentivirus. Our already done experiment by over expressing ERbeta4 on top of all endogenous isoforms is causing more resistance and is thus indicating that ERbeta4 is capable of causing résistance.

  1. Page 6, Figure 4 – In Figure 4, the genes examined in A to D are different. The reviewer thinks that the genes examined in Fig. 4A should also be applied to B to D, and future issues based on the results should be discussed in the Discussion section. In our experience regulation of the two genes analyzed in figure, 4D (Nanog and Oct4) is the most consistent through all cell lines used.

  1. Page 5, line 164- the authors described, “This could be explained by the fact that HCC1806 cell line belongs to a different Lehmann TNBC-subtype - Basal-Like 1, with different molecular signatures”. In the Lehmann TNBC subtype, HCC1806 belongs to a Basal-like 2. The reviewer believes that authors need to elaborate on why HCC1806 could belong to Basal-Like 1in the discussion section. This section if removed.

Minor comment

  1. Page 1, line 15, abstract –the authors used CAS/CRSPR in this section, but the reviewer thinks the correct word is CRISPR/Cas9. This is changed

  1. Page 1, line 16, abstract –the authors used Do-main in this section, but the reviewer thinks the correct word is Domain. This is changed

Reviewer 3 Report

The author analyzed the importance of ERb1 and ERb4 in TNBC cells. However, some issues have to be reviewed.

  1. Define TNBC, ER, PR abbreviation in line 36.
  2. Line 44: “The 5-year overall survival rate in metastatic TNBC (mTNBC) patients was 4–20%…” the correct verb is “is 4-20%
  3. Line 63: is one of the BIGGEST problem
  4. Line 68: add more citations for the HIF1a confirming these results
  5. Line 87: define abbreviations: LBD, NLS 
  6. Supplementary Figure 1: how about the expression of CD44+/CD24-, the hallmark for CSCs in breast cancer?
  7. Introduction is too long
  8. Your goal is “explores further the role of 109 ERβ4 in TNBC”, however your data starts with ERb1 and keeps showing data with ERb1 and ERb4. 
  9. Figure 2B and 2C and 2E - where is the data with CRERB4?
  10. Figure 2D is missing comparing  CRERB4 with the overexpression. If you want to validate you have to show in more cell lines similar to the KO.
  11. How is the expression of ERb4 in these cells? Show PCR data comparing and Western blotting.
  12. Figure 3 - I suggest also add WB analysis 
  13. Line 182 - “we analyzed expression of Yamanaka factors that induce pluripotency in normal cells” (add reference) and define what factors were analyzed.
  14. Figure 4 - I suggest to add flow cytometry for ALDH to confirm PCR results, also the most common used CSCs marker for breast cancer is the expression of CD44+/CD24- and this analysis is missing and should be added.
  15. The citation of supplementary Fig 2-4 is missing in the text.
  16. Figure 5 - I would like to see confirmation of SORE6 results in the other cell lines and how about SORE6 in CRERB4? 
  17. The conclusion has to be reviewed. The study actually focused more in the ERb1 than ERb4.

Author Response

Thank you for good comments and suggestions. Our response is below in italic

Comments and Suggestions for Authors

The author analyzed the importance of ERb1 and ERb4 in TNBC cells. However, some issues have to be reviewed.

  1. Define TNBC, ER, PR abbreviation in line 36.This is done
  2. Line 44: “The 5-year overall survival rate in metastatic TNBC (mTNBC) patients was 4–20%…” the correct verb is “is 4-20% This is changed
  3. Line 63: is one of the BIGGEST problem This is changed
  4. Line 68: add more citations for the HIF1a confirming these results This is done
  5. Line 87: define abbreviations: LBD, NLS This is done
  6. Supplementary Figure 1: how about the expression of CD44+/CD24-, the hallmark for CSCs in breast cancer? According to Wicha there is two types of CSC in breast as mentioned above CD44+/CD24-, which is the mesenchymal one in addition, to this population there is a population of CSC that is expressing ALDH and is more epithelial. Also is present hybrid CSC that are in between the mesenchymal type and the epithelial type. After conferring with Wicha we are convinced that our reporter SORE6 will detect all different CSC populations.
  7. Introduction is too long This have been changed
  8. Your goal is “explores further the role of 109 ERβ4 in TNBC”, however your data starts with ERb1 and keeps showing data with ERb1 and ERb4. The study focuses on a balance between ERbeta1 and ERbeta4 in TNBC.
  9. Figure 2B and 2C and 2E - where is the data with CRERB4? This is not done
  10. Figure 2D is missing comparing CRERB4 with the overexpression. If you want to validate you, have to show in more cell lines similar to the KO. CRERB4 done in SUM159 cells.
  11. How is the expression of ERb4 in these cells? Show PCR data comparing and Western blotting. have been done
  12. Figure 3 - I suggest also add WB analysis Very difficult to do since no good antibody
  13. Line 182 - “we analyzed expression of Yamanaka factors that induce pluripotency in normal cells” (add reference) and define what factors were analyzed. This is done.
  14. Figure 4 - I suggest to add flow cytometry for ALDH to confirm PCR results, also the most common used CSCs marker for breast cancer is the expression of CD44+/CD24- and this analysis is missing and should be added. Is justified not to use since only representing a subpopulation of stem cells (namely the mesenchymal type of stem cells and not the epithelial type of stem cells

  1. The citation of supplementary Fig 2-4 is missing in the text. Figure 5 - I would like to see confirmation of SORE6 results in the other cell lines and how about SORE6 in CRERB4? This is beyond the scope of the study.
  2. The conclusion has to be reviewed. The study actually focused more in the ERb1 than ERb4.  It is Justified to focusing on ERbeta4 since ERbeta4 was found to be the most widely expressed factor in clinical PDX and establisched as the one ERbeta isoform capable of stimulating mammosphere

Round 2

Reviewer 2 Report

The second revised paper seems to be improved and should be worth publishing in this journal. However, the reviewer hopes that change one piece of information (described below).

Minor comment

Page 12, line 265 and page 14, line 293 –the authors used 200 μM in this section, but the reviewer thinks the correct word is 200 µm.

Author Response

Thank you for comment

Comments and Suggestions for Authors

The second revised paper seems to be improved and should be worth publishing in this journal. However, the reviewer hopes that change one piece of information (described below).

Minor comment

Page 12, line 265 and page 14, line 293 –the authors used 200 μM in this section, but the reviewer thinks the correct word is 200 µm. We have corrected this issue and is submitting the corrected manuscript. In addition, the correction have also been done for the supplementary figures.

Reviewer 3 Report

majority of the comments were addressed

Author Response

Thank you for approving changes   Comments and Suggestions for Authors

majority of the comments were addressed